# Associations between Animal Welfare Indicators and Animal-Related Factors of Slaughter Cattle in Austria

**DOI:** 10.3390/ani12050659

**Published:** 2022-03-05

**Authors:** Johann Burgstaller, Thomas Wittek, Nadine Sudhaus-Jörn, Beate Conrady

**Affiliations:** 1Veterinary Service, Municipality of Villach, 9524 Villach, Austria; johann.burgstaller@villach.at; 2Department for Farm Animals and Veterinary Public Health, University of Veterinary Medicine Vienna, 1210 Vienna, Austria; thomas.wittek@vetmeduni.ac.at; 3Institute of Food Quality and Safety, University of Veterinary Medicine Hannover, 30173 Hannover, Germany; nadine.sudhaus-joern@tiho-hannover.de; 4Department of Veterinary and Animal Sciences, University of Copenhagen, 1870 Frederiksberg C, Denmark; 5Complexity Science Hub Vienna, 1080 Vienna, Austria

**Keywords:** abattoir, abomasal ulcers, bovine, cleanliness, contamination, lameness

## Abstract

**Simple Summary:**

The aims of this study were (1) to evaluate the prevalence of lameness, dirtiness of the body surface, and abomasal disorders of slaughter cattle; and (2) to determine the association between these welfare indicators and animal-related factors (e.g., housing type, carcass weight, and transportation and waiting duration of the animals). In contrast to dirtiness (level of contamination of the body surface, also referred to as cleanliness) and the prevalence of abomasal disorders, the determined lameness prevalence was very low. The husbandry of cattle was identified as a significant influencing factor for both the dirtiness and occurrence of abomasal disorders of slaughter cattle.

**Abstract:**

Three cattle welfare indicators (lameness, dirtiness, and abomasal disorders) were evaluated in 412 slaughter cattle in a cross-sectional study in Austria. The aims of this study were (1) to evaluate the prevalence of lameness, dirtiness of slaughter cattle, and abomasal disorders; and (2) to determine the association between these welfare indicators and animal-related factors (e.g., housing type, carcass weight, transportation and waiting duration of the animals). The lameness prevalence was 0.73%, the abomasal disorders prevalence was 52.43%, and 88.59% of all cattle were contaminated. The latter result indicates that the cattle were kept in a dirty environment. The occurrence of abomasal disorders was associated with cattle housing systems (*p* ≤ 0.00) and slaughter weight (*p* = 0.03). The odds for abomasal disorders were 28.0 times higher for cattle housed on slatted flooring compared to cattle kept in a tethered system. The chance for occurrence of abomasal disorders was 3.6 times higher for cattle with a low carcass weight compared to cattle with a high carcass weight. Furthermore, significant associations were found between dirtiness (also referred to as cleanliness or contamination) and husbandry system, sex, and breed. Cattle housed in deep litter boxes had 40.8 times higher odds of being contaminated compared to cattle in a tethered housing system. Cows (odds: 32.9) and heifers (odds: 4.4) had higher odds of being contaminated with feces compared to bulls, whereby female calves (odds: 0.09) and male calves (odds: 0.02) had significantly lower odds of being contaminated. Furthermore, the breeds Brown Swiss (odds: 0.26) and Holstein-Friesian (odds: 0.14) had a significantly lower chance of being contaminated compared to Simmental cattle. Other collected factors, such as production system, transportation duration, life days of the cattle, average daily weight gain, carcass classification, and fat coverage, showed no association with the collected welfare indicators. The study presented here indicates that welfare indicators evaluated for slaughter cattle are suitable to assess cattle welfare, and improvements in husbandry may positively impact both the abomasal physiology and cleanliness of cattle.

## 1. Introduction

More than 70 cattle welfare indicators are described in the literature [1]. These 70 cattle welfare indicators can be assigned to four main categories: morphometric, behavior-specific, physiologic, and meat-quality-affecting indicators. Besides the evaluation of animal welfare indicators on farms, several studies have assessed animal welfare issues in abattoirs [2,3,4,5]. A benefit of using ante- and postmortem indicators, including meat inspection, is the ability to collect data on animal welfare and on food safety from different farms [6]. Data regarding lameness, injuries, emaciation, and cleanliness are considered important animal welfare indicators at the slaughterhouse [7].

Lameness is one of the most important cattle welfare issues [8,9]. Locomotion of cattle can be categorized based on the severity of lameness by a five-point scoring system [10]: (1) clinically normal, i.e., cattle walks and stands with level-back posture; (2) in contrast to score 1, the animal shows an arched-back posture while walking; (3) a moderate lameness is detected if an arched-back posture is observed while the cattle is walking and standing; (4) in contrast to score 3, an arched-back posture is observed all the time; and score (5) indicates a severe lameness characterized by an inability of the cattle to bear weight on one or more of their limbs/feet [10].

Abomasal ulcers and lesions are the most common cause of digestion disorders in cattle of all ages and are important welfare indicators; however, the number of published prevalence studies is low [11,12]. Ulcers lead to pain, loss of production, and death in severe cases [13,14]. Abomasal lesions can be classified, based on distinct variations in clinical signs, between type I (superficial lesions of the abomasal mucosa) and II (deep lesions of the abomasal mucosa), categorized as non-perforating abomasal ulcers, and type III (perforating ulcers with acute, circumscribed peritonitis) and IV (ulcerations with diffuse peritonitis [15]), classified as perforating ulcers [12]. Risk for the occurrence of abomasal ulcers are nutritional factors, destruction of mucosa by high-concentrate feedstuff that decrease pH in the abomasum, mineral imbalance, stress, comorbidities, and medical treatments. For instance, straw is described to damage the mucosa if fed as solid roughage in veal calves [12]. An overview study of identified risk factors for abomasal ulcers was recently published [16].

Fecal contamination of animals represents a hygienic problem during the slaughter process [17]. Dirty cowhides result in higher bacterial loads on concerned carcasses and represent a high risk for cross contamination during the slaughter process [18]. Besides hygiene, animal health, and food safety concern, contaminated animals might indicate poor on-farm management due to dirty flooring and litter, as described previously [19]. A five-step evaluation system is used to assess the level of the cleanliness of cattle developed by the Food Standard Agency (London, UK) [20]. Score 1 includes all clean and dry animals; score 2 incorporates all animals with a slight contamination; score 3 covers all animals with moderate contamination; score 4 indicates a high contamination of the slaughter animal; and score 5 includes all animals with extremely high contamination with feces, resulting in a prohibition of slaughter.

The aims of the present study were (1) to evaluate the prevalence of the animal welfare indicators: lameness, dirtiness (also referred to as cleanliness or contamination) of the body surface, and abomasal ulcers and lesions of slaughter cattle; and (2) to estimate the association between the animal welfare indicators and animal-related factors, such as housing type, production system, transportation and waiting duration, sex, carcass weight, and classification.

## 2. Materials and Methods

The present study was carried out as a cross-sectional study at one slaughterhouse in Austria. The abattoir is EU certified for slaughter of cattle, pigs, and horses. One official veterinarian (J.B.) carried out the sampling at the slaughterhouse on 19 days from 27 July to 30 November 2020. Data were collected regarding (i) lameness, (ii) abomasal disorders, and (iii) dirtiness of slaughter cattle (dependent variables). We focused on these three factors because pre-study observations of the abattoir showed a wide range of dirtiness levels, although no emaciations and injuries were observed. Abomasal disorders were chosen as a factor because the occurrence seems to be far underestimated, measured by a scarcity of scientific studies. In contrast, lameness and dirtiness are commonly used as welfare indicators in scientific studies. Furthermore, these three indicators were also chosen for practical reasons, as one observer scored lameness, dirtiness, and abomasal disorders.

Scoring of lameness was performed on hard ground after the animals arrived at the slaughterhouse based on the five-point scoring system described in [10]. Additionally, all cattle were scored regarding their cleanliness level in the waiting room of the abattoir and after stunning based on the five-point scoring system of the Food Standard Agency [20]. All abomasa were collected and inspected for the presence of lesions. After removing the abomasum from the gastrointestinal tract, it was opened on the side of with greater curvature, and stomach content was washed out. Abomasal lesions were characterized based on the four-point scoring system described in [15,21].

Animal-related metadata (independent variables), such as breed, sex, housing type, transportation and waiting duration, daily weight gain, and production system (see detailed description in Table 1 and Figure 1), were collected for each slaughtered cattle from transport and health certificates, slaughter protocols, and the national cattle database. Information about the type of housing for each individual animal was gathered by interviewing the driver of the cattle transport, who picked cattle up on farms. Table 1 gives an overview of all collected data, the scale and distribution of the collected data, whether the data were included in the statistical analysis, and information about reclassification of some data for the statistical analysis due to low occurrence of some scores. For instance, the five categories of the abomasal lesion score were reduced to two categories (No: free of abomasal lesions; Yes: not free of abomasal lesions). Furthermore, low data variability within the variable “lameness” was determined (see Section 3). Therefore, the statistical analysis was performed for two response variables only, i.e., “abomasal lesions” and “cleanliness level” of the cattle. The “abomasal lesions” and “cleanliness level” of the slaughter cattle were analyzed in two different models due to different scales of the dependent variables: a binary scale of the recorded abomasal lesions (No or Yes) and an ordinal scale of the recorded cleanliness level (i.e., clean, low, medium, and highly contaminated with prohibition of slaughter).

A mixed binomial logistic model (BGLMM) was performed based on Formula (1) to determine the association between the independent variables (*x_i:_* metadata), considered fixed factors (also known as predictor variables), and abomasal lesions alterations (*y_i_*), considered the dependent variable (also known as response variable). The farm and individual cattle were included as random factors (λ_f,c_). A stepwise, backwards factor-selection approach using log-likelihood ratio tests was applied to compare models with different included fixed-factor combinations (with a threshold of 0.05) and to consider only the most relevant fixed factors in the backwards-fitted BGLMM final model [22]. The BGLMM final model was checked for overdispersion and collinearity using the variance inflation factor (VIF). An issue of collinearity exists when VIF is larger than 10, i.e., the variables cannot independently predict the value of the response variables in case of correlation between predictor variables [23]. To determine the goodness of fit and the predictive accuracy of the BGLMM model, a data partition was conducted in the training (60% of the dataset) and testing dataset (40% of the dataset). In this context, a confusion-matrix, representing the matches and mismatches between predictions and actual results, was calculated to determine the accuracy, sensitivity, and specificity, as well as the positive and negative predictive values for abomasal lesions.
(1)$$y_{i}=\{\begin{array}{c} 1\;\mathrm{if}\;\mathrm{the}\;i-\mathrm{th}\;\mathrm{cattle}\;\mathrm{has}\;\mathrm{abomasal}\;\mathrm{lesions}\\ 0\;\mathrm{otherwise} \end{array}$$
*y**_i_* = *ß*_0_ + Σ*ß_i_x_i_* + λ_f,c_ + *E*(2)

To identify associations between “cleanliness level” (*Y*) of the cattle and the independent variables (i.e., animal-related metadata as predictor variable, also known as fixed factors), a cumulative link mixed model (CLMM) with adaptive Gauss–Hermite quadrature approximation was used. Because other farm specific herd management factors may also contribute to the cleanliness level of cattle, we considered the “farm (*n* = 97)” (*b_f_*) and the “individual cattle identification number (*n* = 412)” (*b_c_*) as random effects in the CLMM [24] based on the assumption of normally distributed random effects (see formula 2). The odds for the clean, low, medium, and high contamination of the slaughtered cattle were modelled by the CLMM based on formula 2, where θ*_j_* describes the flexible threshold for category *j*, *x_i_* specifies the fixed meta factors, *i* is the index of all observations, and *j* = 1, …, J is the index of the response categories (J = 4; clean, low, medium, and high contamination). In the first step, we calculated a full initial CLMM model (i.e., considered all fixed meta factors, *x_i_*, in Table 1) by subsequent application of a drop function with a Chi-squared (likelihood ratio) test to include the significant *x_i_* fixed factors in the final model [25]. The correlation between the final predictor variables in the model were assessed with Carmer’s V (φc) (φc: 0 = no correlation/association; 1 = high correlation/association). In the second step, we calculated the random effects, *b_f_* and *b_c_*, via conditional modes with 95% confidence intervals based on the conditional variance and the expected probability (including 95% confidence intervals) for the cleanliness level using the ggpredict function (see Appendix A).
logit(*P*(*Yi* ≤ *j*)) = θ*_j_* − *ß*_1_*x_i_*_,1_ − … − *ß**_n_*,*x_i_*_,*n*_ − *u_i_b_f_*_,*c*_;(3)
*b*_*f*,*r*_~*N*(0, σ^2^*_f_*) *r* =1…97; *b_c_*_,*m*_~*N*(0, σ^2^*_c_*) *m* = 1…412(4)

In both models, the significance level was set to *p* < 0.05. The models were implemented using the packages ‘ordinal’, ‘ggeffects’, ‘lme4’, ‘car’, ‘caret’, ‘lattice’, ‘ROCR’, ‘pROC’, and ‘LMERConvenienceFunctions’ in the R (Version 4.0.5) statistical computing environment.

## 3. Results

Data were collected from 412 slaughtered cattle representing 97 cattle farms. The lameness prevalence in the study presented here was 0.73% (*n* = 3) of all slaughter cattle (i.e., including slight (score 2) and moderate (score 3) lameness). No cattle were identified with severe lameness (scores 4 and 5). Thus, the majority of the analyzed slaughter cattle had no lameness (99.27%; score 1 (*n* = 409)). The frequency of deep abomasal ulcers was low (i.e., 7.78% (*n* = 32; type 2) and 0.24% (*n* = 1; type 3), respectively) compared to cattle free of lesions (47.81% (*n* = 197; type 0)) and cattle with superficial ulcers of the abomasal mucosa (44.17% (*n* = 182; type 1)). Types 1, 2, and 3 were cumulated into one group, and abomasal lesions were categorized as a binary variable for statistical analysis (no: free of abomasal mucosa lesions and ulcers (type 0); yes: not free of abomasal mucosa lesions and ulcers (types 1, 2, and 3)). Overall, the prevalence of lesions of the abomasal mucosa was 52.43%.

In total 11.40% of the cattle were assigned to cleanliness level 1 (clean), 60.92% to level 2 (low contamination), 22.83% to level 3 (medium contamination), and 4.85% to level 4 (high contamination). No slaughtered cattle were assigned to the level 5 (slaughtering would have been prohibited due to hygienic deficiencies; see Table 1). In total, the contamination prevalence of the slaughter cattle was 88.59%.

Figure 1 illustrates the scorings of the slaughter cattle regarding “lesions of the abomasal mucosa” and “cleanliness” at the slaughterhouse. The average transportation and waiting duration of the slaughter cattle was 7.03 h (min: 0.05 h; max: 29.43 h). The average live days was 393 days (min: 60 days, max: 1981 days), the mean carcass weight was 262 kg (min: 58 kg, max: 762 kg), and the average daily weight gain of slaughter cattle was 0.66 kg (min: 0.16 kg, max: 1.37 kg). These animal-related data were recategorized for statistical analysis as low, medium, or high. Table 1 shows the frequency distribution of all collected data of the slaughter cattle.

After the backwards factor-selection approach and after excluding “sex” as a fixed factor due to collinearity with slaughter weight, the final BGLMM model included a total of three fixed factors: “housing type”, “carcass classification”, and “slaughter weight”. The frequency of these three factors in relation to the binary abomasal classification is shown in Figure 2a. Significant associations were determined between abomasal lesions and “housing type” and “slaughter weight”, respectively. No association was identified between abomasal lesion and “carcass classification” (Table 2). The odds for occurrence of abomasal lesions were 28 times higher for cattle housed with slatted flooring compared to a tethered housing system. The odds for abomasal mucosa lesions compared to no occurrence of abomasal mucosa lesions were 3.69 times higher for animals with low slaughter weight class (<150 kg) compared to cattle with a high slaughter weight (>300 kg). The mean accuracy of the model in cross validation was 68.0% (see ROC analysis, Appendix A). The latter model result indicated that 32.0% of the abomasal lesion cases would be misclassified by using these three predictor factors.

After applying the dropping function, the final CLMM included three fixed factors, i.e., “housing type”, “sex”, and “breed”. No correlation between these three factors was identified (i.e., φc: 0.47 between “housing type” and “sex”; φc = 0.38 “housing type” vs. “breed”; and φc = 0.30 “sex” vs. “breed”). All three factors were identified as significant influencing factors on the cleanliness level. The relative frequency of these three factors stratified by “cleanliness level” is shown in Figure 2b. Considering the “housing type” of the cattle, “deep litter boxes” (odds: 40.82) and “calves in group housing on straw bedding” (odds: 7.04) had significantly higher odds of contamination compared to cattle kept in a “tethered housing system”. Table 3 shows that the odds for contamination were significantly lower for “Brown Swiss” (odds: 0.26) and “Holstein-Friesian” (odds: 0.14) compared to “Simmental” cattle. The odds for contamination were significantly higher for “cows” (odds: 32.95) and “heifers” (odds: 4.40) and lower for “female calves” (odds: 0.09) and “male calves” (odds: 0.22) compared to “bulls”. The estimated variance components for the random effects, farm (σ2f) and cow (σ2c), were 0.1 and 0.0, respectively (Appendix A shows the associated random effect of the single farms). The probability for a specific contamination level stratified by “housing type”, “sex”, and “breed” is shown in Appendix A. The expected probability for clean cattle was more than 50% for both female calves and male calves across all housing types and breeds, whereas the probability was less than 1% for “high” contaminations.

## 4. Discussion

Animal welfare is a growing demand of the modern consumer. People ask for food produced from animals kept under animal-friendly and fair conditions, understood as the treatment, transport, and slaughter of animals without causing pain, suffering, lesions or severe fear of the animals [24]. To improve the situation for the animals, two approaches are common today. The first is to evaluate the environment and husbandry, and the second is to focus on the animal itself by observing the impact of the environment on behavior, wellbeing, and health. Animal welfare indicators describe how animals cope with their production and husbandry system. This study describes three animal welfare indicators in slaughter cattle. One official veterinarian scored all animals regarding these factors (lameness, contamination level, and abomasal lesions); therefore, we assume no interobserver bias in the present study. Data were collected at one abattoir in Austria. The animals came from 97 holdings; therefore, a huge variety of farm individual factors had to be taken into account. Animal and farm-specific data were collected from transport certificates, the national cattle database, and by interviewing the transporter of the individual animals. Animal and farm-specific data, including findings from official meat inspection and data from the independent meat classification company, were defined as possible risk factors for the three predefined animal welfare indicators. The study results show a significant association between contamination level and husbandry system, sex, and breed, as well as significant associations between abomasal lesions and housing type and slaughter weight.

The first part of the study describes the prevalence of lameness, contamination, and abomasal lesions in slaughter cattle. A total lameness prevalence of 0.7% was observed, which can be compared to the results of studies by Fjeldaas et al. [28] describing 1.1% lame cattle in suckler herds, which is lower than the 2.0% documented by the University of Nebraska for US feedlot cattle [29]. The observers’ expertise in scoring lame cattle was high [24,30], so we assume that even slightly lame animals could be detected. This point is crucial, as farmers often underestimate the prevalence of lameness in their herds [31]. Slaughter animals in this study were mainly young bulls and calves, a group of animals that is far less affected by lameness compared to dairy or cull cows [32,33]. An on-farm lameness scoring before loading would be helpful to determine the prevalence in a more accurate way; however, highly lame animals would not arrive at the slaughterhouse because of legal restrictions concerning transport of these animals. Contamination of the slaughter animals’ bodies was scored by using a five-level system. The results for cleanliness are in line with those of another recent study from Austria [17] and a study from Serbia [34] and show that slaughter animals are often kept in a dirty and therefore unhygienic environment. This indicates strong deficits in animal husbandry and management on farms [19]. The overall prevalence of abomasal lesions was 52.4%, whereas 60.3% of the bulls, 43.1% of female calves, and 55.6% of male calves had abomasal lesions in the study at hand. Similar results were reported by Hund et al. [11], although higher prevalences were reported, ranging from 70.0 to 93.0% [16]. In total, 26.8% of slaughtered heifers in this study showed abomasal lesions, almost the same percentage as that reported by Jensen et al. [35] of 24.9%. Transport and waiting time from loading on the farm to stunning was calculated. The time of loading has to be documented by the farmer who accompanies the cattle to their destination. The time of stunning was documented by the observer of this study (JB). The average transportation and waiting duration was 7.03 h (min: 0.05 h; max: 29.43 h). The average transport and waiting time for slaughter cattle transported by the farmer himself was 2.14 h, compared to 7.45 h for cattle transported by an external company. The duration of slaughter animal transport within Austria is limited to 8 h by national regulations [36]. The total transport time could not be determined because some of the animals had been unloaded at collection points during their journey. Some of the animals were unloaded at the slaughterhouse during the night before slaughter. Nevertheless, the total transport and waiting time of slaughter animals from loading to stunning as reported in this paper is long; however, in this study, no association was determined with respect to the considered welfare indicators. Slaughter cattle therefore have to cope with stress, new environments, and unknown situations during that time. Preferably, slaughter cattle should be transported in small groups by farmers to avoid or mitigate negative experiences.

The second part of the study, i.e., model results on the occurrence of alterations of the abomasal mucosa, showed that cattle kept on slatted flooring had a significantly higher chance of developing ulcers compared to cattle kept in tie stalls. This fact might show that cattle on slatted flooring have limited space to sidestep in their boxes and develop more stress, even though daily weight gain and production data are not affected. Slaughter cattle with lower carcass weight showed significantly more alterations of abomasal lesions. Therefore, we assume that either cattle with abomasal ulcers are slaughtered earlier when they weigh less due to lower production performance or abomasal ulcers recover during later fattening periods. Furthermore, cattle housed on deep litter flooring have the highest chance of being the dirtiest. These results are in line with those of a study from Italy comparing welfare and cleanliness of finishing bulls [37]. The reason for this finding might be that most deep-litter flooring systems are managed poorly, i.e., are not well cleaned. Sex also turned out to have a significant influence on cleanliness. This result was biased by the fact that mainly bulls and calves were observed in this study, with only a few heifers and cows. Besides that, bulls were mainly kept on slatted flooring, whereas heifers were kept on deep-litter flooring. The factor of breed had a significant influence on cleanliness as well, showing that Brown Swiss and Holstein-Friesian had a higher chance of being cleaner. This effect was mainly influenced by the lower age of Brown Swiss and Holstein-Friesian calves compared to Simmental.

## 5. Conclusions

In contrast to dirtiness and the prevalence of abomasal disorders, the determined lameness prevalence was very low. The husbandry of cattle was identified as a significant influencing factor for both the dirtiness and occurrence of abomasal disorders of slaughter cattle. Gathering data on animal welfare indicators at slaughterhouses is useful to determine potential influencing factors on the side of production.

## Figures and Tables

**Figure 1 animals-12-00659-f001:**
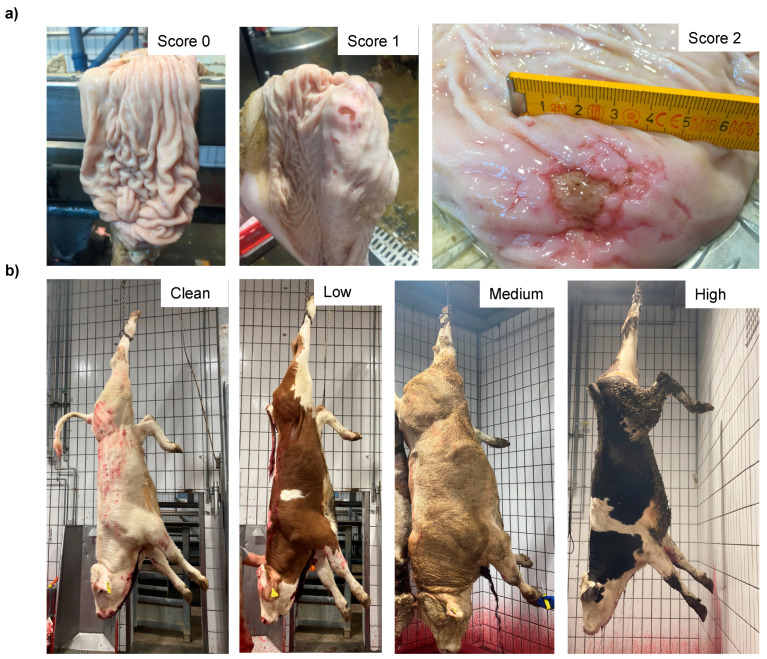
Illustration of the different scales of the dependent variables (**a**) abomasal lesions (i.e., 0 = normal abomasal mucosa (free); 1 = superficial lesions of the abomasal mucosa; 2 = deep lesions of the abomasal mucosa) and (**b**) cleanliness level (i.e., clean; low contamination; medium contamination; high contamination). Pictures were taken by the first author during the study.

**Figure 2 animals-12-00659-f002:**
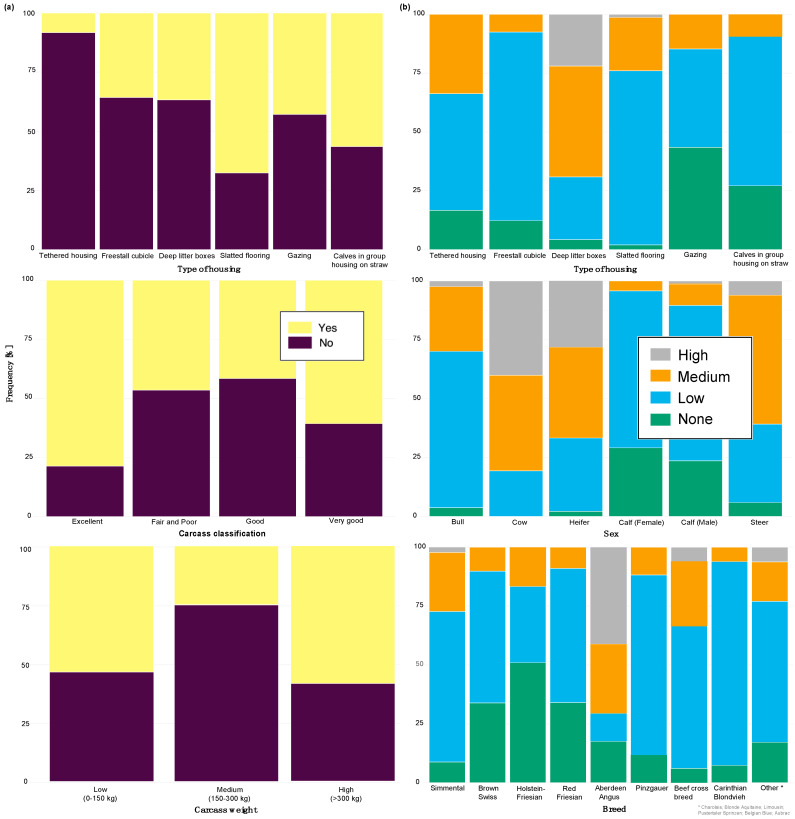
Frequency of recorded (**a**) abomasal mucosa alterations (Yes = presence of abomasal mucosa lesions; No = no presence of abomasal mucosa lesions) stratified by fixed factors considered in the final model, i.e., type of housing, carcass classification, and carcass weight, and (**b**) cleanliness level stratified by fixed factors considered in the final model, i.e., type of housing, sex and breed.

**Table 1 animals-12-00659-t001:** Overview of scores, classifications, and distribution of the collected animal welfare and animal-related metadata of slaughter cattle.

Category	Description	Scoring System	(a) Classification of Data (b) Category Included in the Model (Yes; No)	Absolute Frequency of Collected Data
Lameness	Scoring was performed on hard ground after the animals arrived at the slaughterhouse based on the scoring system by Sprecher [10]	1 = sound 2 = slight lameness 3 = moderate lameness 4 = high lameness (i.e., movement was only possible with claw tip) 5 = severe lameness (i.e., cattle did not touch the ground with one leg)	(a) 0 = free (covered score 1) 1 = not free (covered scores 2–5) (b) No; although reclassified, still low data variability	In total, 99.27% (*n* = 409) of all cattle were not lame. Only three cattle were identified with lameness.
Abomasal lesions	Scoring of abomasal lesions was performed after the abomasum was cut from the convolute, opened on the side of the greater curvature, and the stomach contents were washed out [15,21]	0 = normal abomasal mucosa (free) 1 = superficial lesions of the abomasal mucosa 2 = deep lesions of the abomasal mucosa 3 = ulceration of the abomasum, which triggered local inflammatory reactions 4 = ulceration of the abomasum with generalized peritonitis	(a) 0 = No (covered score 0) 1 = Yes (covered scores 1–4) (b) Yes, as dependent variable (response variable)	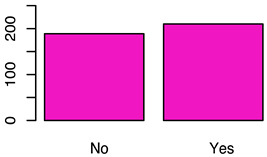
Cleanliness	Scoring of the cleanliness of the cattle was carried out in the waiting room and after stunning based on the Clean Livestock Policy of the British Meat Hygiene Service [20]	1 = clean 2 = low contamination 3 = medium contamination 4 = high contamination 5 = slaughtering is prohibited due to hygienic deficiencies	(a) see (column: score system) (b) Yes, as dependent variable (response variable). Category 5 was not identified in this study	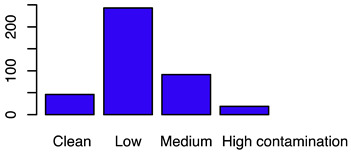
Pathological findings	The presence of pneumonia, kidney alterations, liver flukes, abscesses, and pleuritis was recorded	0 = absent 1 = present	(a) see (column: score system) (b) No, due to low number of detected pathological issues	Few pathological findings were recorded for slaughtered cattle pneumonia (0.04% of all cattle), kidney (0.00%), liver flukes (0.01%), abscess (0.01%), and pleuritis (0.00%) (N = 412)
Housing type	The type of housing was collected by interviewing the transporter who picked up the animals	Tethered Freestall cubicle Deep litter boxes Slatted flooring Grazing Calves in group housing on straw bedding	(a) see (column: score system) (b) Yes, as independent variable (fixed-factor)	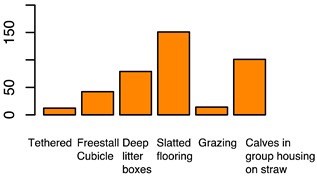
Breed	The breed was collected from the transportation certification and/or from the national cattle databases based on the ear-tag number	Simmental Brown Swiss Holstein-Friesian Red Friesian Aberdeen Angus Pinzgauer Beef cross breed Carinthian Blondvieh Charolais Blonde Aquitaine Limousin Pustertaler Sprinzen Belgian Blue Aubrac	(a) Simmental (SI) Brown Swiss (BS) Holstein-Friesian (HF) Red Friesian (RF) Aberdeen Angus (AA) Pinzgauer (P) Beef cross breed (BCB) Carinthian Blondvieh (CB) Other (O) (i.e., Charolais, Blonde Aquitaine, Limousin, Pustertaler Sprinzen, Belgian Blue, and Aubrac) (b) Yes, as independent variable (fixed-factor)	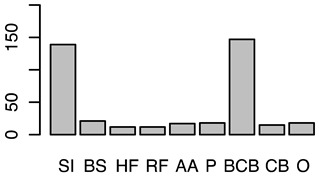
Sex	The sex was identified by the observer and rechecked against the transportation certification and the national cattle databases	Bull Cow Heifer Calf (female) Calf (male) Steer	(a) see (column: score system) (b) Yes, as independent variable (fixed-factor)	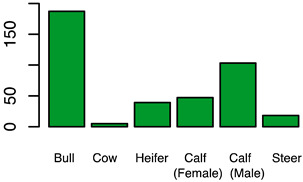
Production system	The type of production system was collected from animal transport certification	Commercial Organic	(a) see (column: score system) (b) Yes, as independent variable (fixed-factor)	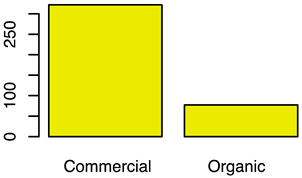
Who transported?	The type of transportation was collected from the transportation certification	External company Farmer	(a) see (column: score system) (b) Yes, as independent variable (fixed-factor)	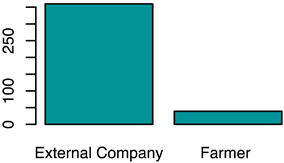
Transportation and waiting duration	Transportation and waiting duration was calculated from transport certification (loading time) and time of stunning	Numeric	(a) Low ≤ 4 h Medium = 4−8 h High ≥ 8 h (b) Yes, as independent variable (fixed-factor)	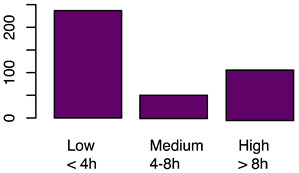
Date of birth	Date of birth was recorded from transport certification and rechecked against the national cattle database	Numeric	(a) – (b) No	Not applicable as diagram
Life days of the cattle	Life days were calculated based on date of birth and day of slaughter	Numeric	(a) Low ≤ 300 days Medium = 300−600 days High ≥ 600 days b) Yes, as independent variable (fixed-factor)	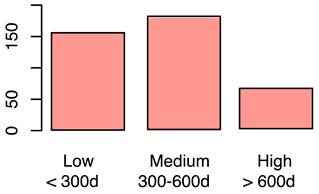
Carcass weight	Carcass weight was collected after slaughtering (warm weight)	Numeric	(a) Low ≤ 150 kg Medium = 150−300 kg High ≥ 300 kg (b) Yes, as independent variable (fixed-factor)	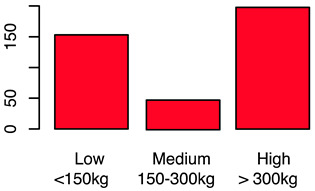
Average daily weight gain	Average daily weight gain was calculated based on the slaughter weight minus an assumed slaughter weight on the first day of 20 kg (40 kg life weight), divided by the days of life	Numeric	(a) Low ≤ 0.4 kg Medium = 0.4−0.8 kg High ≥ 0.8 kg (b) Yes, as independent variable (fixed-factor)	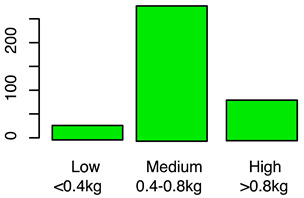
Carcass classification	Carcass classification was performed by an independent classification company using the EUROP classification system according to EU regulation [26]	E = excellent muscle development U = very Good R = good O = fair P = poor	(a) E = excellent U and R = very good/good O = fair P = poor (b) Yes, as independent variable (fixed-factor) N.B. O and P were summarized as one category due to a single record in category P	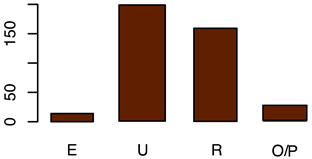
Fat coverage	Fat coverage classification was performed by an independent classification company according to EU regulation [26]	1 = none 2 = slight 3 = average 4 = high 5 = very high	(a) see (column: score system) (b) Yes, as independent variable (fixed-factor). N.B. (very) high fat coverage was not detected (levels 4 and 5).	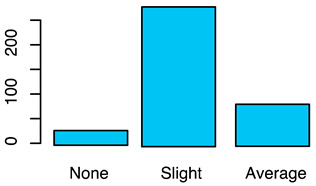
Color classification of the meat	Color classification of the meat was performed by an independent classification company based on the color scheme of AgrarMarkt Austria [27]	1−4 = white–light rose 5−8 = rose−dark	(a) see (column: score system) (b) No	2/3 of the data were missing, as the color of meat is only determined for calves
Farm registration number	Farm registration number was collected from the transportation certification	Factor	(a) see (column: score system) (b) Yes, as random factor	Not applicable as diagram
Ear-tag number	Ear-tag numbers of the cattle were collected from the transportation certification	Factor	(a) see (column: score system) (b) Yes, as random factor	Not applicable as diagram

**Table 2 animals-12-00659-t002:** Estimated influencing fixed factors (including associated odds) for the abomasal mucosa lesions of the slaughtered cattle compared to the intercept.

Fixed Effect ^1^	Estimated Coefficient	Odds (95% Confidence Intervals)	*p* Values
Freestall cubicles	0.77	2.17 (0.20–23.20)	0.521
Deep litter flooring	1.43	4.18 (0.45–38.30)	0.204
Slatted flooring	3.33	28.00 (2.68–292.0)	0.005 **
Grazing	1.13	3.12 (0.23–40.70)	0.385
Calves in group housing on straw	1.43	4.18 (0.39–44.70)	0.236
Carcass classification: U	−0.62	0.53 (0.13–2.06)	0.361
Carcass classification: R	−1.23	0.29 (0.07–1.19)	0.084
Carcass classification: O/P	−1.37	0.25 (0.04–1.31)	0.101
Carcass weight: low (<150 kg)	1.30	3.69 (1.21–11.30)	0.022 *
Carcass weight: medium (150–300 kg)	−0.00	0.99 (0.38–2.60)	0.991

Significance codes: ** ≤0.01; * ≤0.05; CI = confidence intervals. ^1^ The intercept includes housing system = tethered; carcass classification = E (excellent); carcass weight = high (>300 kg).

**Table 3 animals-12-00659-t003:** Estimated significant influencing fixed factors (including associated odds) for the contamination of the slaughtered cattle compared to the intercept.

Fixed Effect	Estimated Coefficient	Odds (95% Confidence Intervals)	*p* Values
Housing type ^1^			
Freestall cubicles	1.16	3.21 (0.58–17.57)	0.177
Deep litter flooring	3.70	40.82 (7.89–211.12)	<0.000 ***
Slatted flooring	1.49	4.46 (0.87–22.89)	0.072
Grazing	−0.07	0.93 (0.09–8.68)	0.950
Calves in group housing on straw	1.95	7.04 (1.24–39.87)	0.027 *
Sex ^1^			
Cow	3.49	32.95 (2.94–368.29)	<0.000 **
Heifer	1.48	4.40 (1.55–12.52)	<0.000 **
Calf (female)	−2.38	0.09 (0.02–0.35)	<0.000 ***
Calf (male)	−1.49	0.22 (0.06–0.72)	0.012 *
Steer	0.07	1.07 (0.29–3.86)	0.911
Breed ^1^			
Brown Swiss	−1.32	0.26 (0.07–0.93)	0.038 *
Holstein-Friesian	−1.94	0.14 (0.03–0.63)	0.010 *
Red Friesian	−1.24	0.28 (0.07–1.12)	0.073
Aberdeen Angus	1.28	3.59 (0.89−14.44)	0.070
Pinzgauer	−0.05	0.94 (0.26–3.43)	0.934
Beef cross breed	0.00	1.01 (0.57–1.79)	0.975
Carinthian Blondvieh	1.31	3.73 (0.91–15.17)	0.065
Other breeds ^2^	−0.17	0.84 (0.25–2.79)	0.777

Significance codes: *** ≤0.001; ** ≤0.01; * ≤0.05; CI = confidence intervals; Intercept: ^1^ = Housing type = as reference the tethered housing type was used; sex = as reference the bulls were used; breed= as reference the Simmental was used. ^2^ = Charolais; Blonde Aquitaine; Limousin; Pustertaler Sprinzen; Belgian Blue; Aubrac.

## Data Availability

The digital datasets and analyses of the present study are available from the corresponding author upon request.

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
