# Peer review of "Associations between Animal Welfare Indicators and Animal-Related Factors of Slaughter Cattle in Austria"

_animals, 2022, doi:10.3390/ani12050659_

Round 1

Reviewer 1 Report

ID: Animals-1605345: Associations between Animal Welfare Indicators and Animal Related Factors of Slaughter Cattle in Austria

The study evaluated the prevalence of lameness, cleanliness, and abomasal ulcers and their association with some indicators of animal welfare. The research is of scientific interest and its results are useful to cattle producers and slaughterhouse.

The manuscript is well structured, easy to read and understand. Presentation and discusion of results are adequate. However, there are some minimum recommendations that must be met before acceptance for publication.

Results section:

In tables 2 and 3, it is recommended to place the P-values with three digits

In the footnote of table 2, indicate what mean "*" and "**" (prob level), as it was indicated in table 3.

The text of lines 250-258 (One official veterinarian....until...pre-defined animal welfare indicators.) are methodological aspects, therefore it should be included in the materials and methods section. Maybe after line 93.

Conclusions:

Although the Conclusion is accurate and correct, it may be necessary to have a broader context of it. For example, mention that there was a high prevalence of abomasal ulcers and superficial dirt on the animal's body, and that these welfare indicators were associated with factors such as the housing type, sex, breed and animal weight.

Author Response

Dear Reviewer 1, we have incorporated your comments in the revised version of our paper. Thank you for your time and valuable improvements. 

Comments Reviewer 1: 

The manuscript is well structured, easy to read and understand. Presentation and discussion of results are adequate. However, there are some minimum recommendations that must be met before acceptance for publication.

Results section:

In tables 2 and 3, it is recommended to place the P-values with three digits

  • Done as suggested.

In the footnote of table 2, indicate what mean "*" and "**" (prob level), as it was indicated in table 3.

  • Thank you for your excellent comment. I have now included the same legend regarding the significance level from Table 3 for Table 2.

The text of lines 250-258 (One official veterinarian....until...pre-defined animal welfare indicators.) are methodological aspects, therefore it should be included in the materials and methods section. Maybe after line 93.

  • This information is already in the material section (please see line 90). But we have rewritten the sentence that it fits more in the discussion section therefore we assume no interobserver bias had to be considered in the present study

Conclusions:

Although the Conclusion is accurate and correct, it may be necessary to have a broader context of it. For example, mention that there was a high prevalence of abomasal ulcers and superficial dirt on the animal's body, and that these welfare indicators were associated with factors such as the housing type, sex, breed and animal weight.

  • Thank you for your comment. Additionally, we have now included the following sentence in the conclusion section: In contrast to dirtiness and abomasal disorders prevalence, the determined lameness prevalence was very low. The husbandry of cattle was identified as a significant influencing factor for both the dirtiness and occurrence of abomasal disorders of slaughter cattle.

Reviewer 2 Report

Dear Authors

Thank you for this very interesting and important report of your research at the slaughterhouse.

General comments: in the simple summary and in the abstract it would be easier to understand if you would describe the three response variables negatively, because you are talking about “prevalence of…” which sounds little strange if you refer to health and cleanliness. I recommend to write: … to evaluate the prevalence of lameness, dirtyness, and abomasal disorders of slaughter cattle,…

You are talking about abomasal health, abomasal lesions, and abomasal ulcers, abomasal disorders, later on abomasal mucosa alterations. In the introduction part some of these expressions are well defined. But when reading the summary and the abstract, the definitions are not known. Therefore, abomasal disorders is the best expression at that stage. Please use that one for a better overview.

In the introduction part you are explaining the chosen welfare indicators. But could you please state why you chose those 3 and not emaciation and injuries or others like stress indicators. And could you also say where else these indicators are usually used? You are describing the scores you used already in this part. I think they could – as they are – be moved to the methods part. This would not be the case if you would describe different possible scoring systems, here. This would be interesting.

In the methods part you are not exactly describing all the methods and scores you used. You are describing them in table 1, which is placed in the results part. But this table, as well as Figure 1 belong clearly to the methods part! In the last column of table 1 you show the distribution of the different scores in your observations and I guess you thought that those are results. This is true, but they were actually used to calculate the statistical models which are also described in the methods part and the greatest part of table 1 is clearly on methods. Therefore, I strongly recommend to move Table 1 and Figure 1 to the methods part.

In the methods part you are always talking about the “cleanliness score”. But in the results part you are showing “contamination levels” (Figure 2). This should be consistent throughout the paper.

Statistics: did you calculate any interactions? I wondered whether there could have been an interaction between breed and housing system.

The results part contains a lot of methods. Also lines 163 to 173 and lines 179 to186 should be moved to the methods part. On the other hand, there are aspects lacking in the results part, which appear only in the discussion part: lines 287 – 289 and lines 293 -295 describe interesting results (that transport- and waiting time is lower when farmers bring their animals to the slaughterhouse and that those parameters don’t have an impact on the observed welfare indicators). Those results have to be mentioned in the results part and not only in the discussion part.

The conclusions part is too short. Please give your recommendations resulting from your study, here.

General remark on language: I found too many spelling and grammar mistakes (see below): those could have been avoided if a person with good English skills would have corrected the manuscript before submitting!

Details:

Line 28: replace housing by housed and flooring by floor

Line 29: replace keep by kept and delete husbandry

Line 31: replace “a significant association was “ by “significant associations were”

Line 32: delete “a”

Line 35: delete “a”

Line 40: replace indicated by indicates

Line 41: replace “the husbandry of cattle might” by “husbandry may”

Line 47: replace “separated into” by “assigned to”

Line 49: replace “on abattoirs” by “in abattoirs”

Line 54: replace issue by issues

Line 55: replace “severity by a five-point lameness scoring system” by severity of lameness by a five-point scoring system”

Line 56: replace “clinical normal cattle i.e.” by clinically normal, i.e.” and replace walk by walks

Line 58: replace “was defined” by “is detected”

Line 60: replace severely by severe

Line 62: replace “ulcers” by “ulcers and lesions” and replace “for ingestion” by “of digestion disorders” and add “are important welfare indicators” after ages

Line 68: delete including

Line 69: delete factors

Line 70: please explain which feedstuffs! Ant replace treatment by treatments

Lines 75-76: replace “might be an indicator for” by “might indicate”

Line 77: replace “five-stage” by “five-step”

Line 79: replace incorporate by incorporates and slightly by slight

Line 81: delete such

Line 84: replace cleanliness by dirtiness and replace ulcers by ulcers and lesions

Line 87: replace carcasses by carcass and add sex after weight

Line 89: replace presented by present

Line 90: replace slaughtering by the slaughter

Line 91: add “the” between out and sampling

Line 93: replace health by disorders and cleanliness by dirtiness

Line 97: add “the” between on and five-point

Line 102: move housing type after sex

Line 103: replace detail by detailed. Here you are referring to table 1, but this one should be in the methods part!

Line 110: add “the” between of and abomasal. And replace was by were. And delete the separator between two and categories

Line 113: add “,only” after variables.

Line 115: replace scale by scales

Line 116: replace vs by or. Add “an” between “and” and “ordinal”

Line 117: add “the” between of and recorded

Lines 121/122: replace mucosa alterations by lesions

Lines 128-130: delete this sentence explaining reference 23, it is not necessary

Line 140: replace variable by variables

Line 143: replace “to cleanliness level of the cattle” by “to the cleanliness level of cattle”

Line 144: replace effect by effects

Line 145: add “the” between on and assumption

Lines 181-183: please rephrase this sentence to: The average live days were 393 days (min: 60d, max: 1981d), the mean carcass weights were: 262 kg (min: 58 kg, max: 762 kg), and averge daily weight gains of slaughter cattle were 0.66 kg (min: 0.16kg, max: 1.37 kg).

Line 185: replace show by shows

Line 188: replace scale by scales and variable by variables and health by lesions.

Line 190: the medium contamination can’t be seen on the picture. Don’t you have a better one?

Table 1: replace abomasal health by abomasal lesions. Replace Score system by scoring system (many times) Lameness score 3= moderate lameness (instead of moderate lame). Replace commercial by organic and carcasses by carcass.

In the methods part “pathological findings” as well as transportation and waiting duration and production system (replace “commercial” by “organic”!) and daily weight gain had not been mentioned so far. Please add them where you are talking about your observations.

Why do you write “NA” concerning frequency of collected data in farm number, ear tag number and date of birth? Should you not rather write: “all collected”?

Line 195: replace finale by final

Line 196: replace carcasses by carcass

Line 197: I don’t understand the word stratified here

Line 200: replace carcasses by carcass

Line 201: replace was by were

Line 203: replace was by were

Line 209: replace alterations by lesions

Figure 2: replace carcasses by carcass

Line 213: replace present by presence, replace alterations by lesions, replace not present by no presence

Line 224: replace keep by kept. Add system after housing

Lines 224/225: delete of the cattle

Line 235: replace cleanliness by contamination level

Lines 243/244: replace “Furthermore, the” by “Those are understood as”. Delete “has to be done in a way that does not”

Line 245: replace cause by “without causing”. Replace fear to by fear of

Line 246: please add a reference to this! Replace First by The first one and delete “the” before environment

Line 247: replace and second by and the second one. Replace to see by observing

Line 248: delete “the” between on and behavior. Delete “of the animal”

Line 251: replace cleanliness by contamination level and abomasal health by abomasal lesions

Line 252: replace was gathered by were collected

Line 254: replace was by were

Line 259: replace cleanliness by contamination

Line 260: replace “a significant association” by “significant associations”

Line 262: replace cleanliness by contaminations

Line 263: replace ulcers by lesions.Replace seen by observed

Line 267: replace have been by could be (or were)

Line 270: replace the by a

Line 272: do you have a reference for that? Is it really possible that animals don’t show lameness if they are lame?

Lines 273 and 274: replace cleanliness by contamination

Line 275: make a point after Serbia (34)

Line 276: replace but by They. Add “often” between  that and slaughter

Line 280: replace have been by were

Line 281: replace have also been by were

Line 291: replace determinated by determined

Line 292: delete to slaughter

Line 297: replace preferable by preferably,

Line 298: replace experience by experiences

Line 300: replace have by had

Line 303: replace is by are

Line 304: replace significant by significantly. Replace alterations by lesions

Line 307: replace period by periods

Line 310: you say they are managed poorly. Don’t you want to add some details (not enough straw, not well cleaned, too many animals?)

Line 313: so, is there an interaction between sex and husbandry?

Line 314: add “a” between had and significant

Line 320: add “the” between on and side

Line 338: replace transporter by transporters

Author Response

Dear Reviewer 2,

thank you for your valuable comments and time. We have now incorporated your comments in the revised version of our paper.

Dear Authors

Thank you for this very interesting and important report of your research at the slaughterhouse.

General comments: in the simple summary and in the abstract it would be easier to understand if you would describe the three response variables negatively, because you are talking about “prevalence of…” which sounds little strange if you refer to health and cleanliness. I recommend to write: … to evaluate the prevalence of lameness, dirtyness, and abomasal disorders of slaughter cattle,…

  • Thank you for your great comment. We have now replaced „cleanliness“ with „dirtiness“ and „abomasal health“ with „abomasal disorders“ throughout the simple summary and in the abstract.

You are talking about abomasal health, abomasal lesions, and abomasal ulcers, abomasal disorders, later on abomasal mucosa alterations. In the introduction part some of these expressions are well defined. But when reading the summary and the abstract, the definitions are not known. Therefore, abomasal disorders is the best expression at that stage. Please use that one for a better overview.

  • Thank you for your comment. We have now replaced the different terms for "abomasal" with "abomasal disorders“ throughout the summary and the abstract.

In the introduction part you are explaining the chosen welfare indicators. But could you please state why you chose those 3 and not emaciation and injuries or others like stress indicators. And could you also say where else these indicators are usually used? You are describing the scores you used already in this part. I think they could – as they are – be moved to the methods part. This would not be the case if you would describe different possible scoring systems, here. This would be interesting.

  • Thank you for your comment. We have now included for the reader of the journal the reason why we chose the three indicators in the material section (please see lines 146-152): We focused on these three factors because pre-study observations on the abattoir showed a wide range of dirtiness levels but no emaciations and injuries were observed. Abomasal disorders were chosen as a factor because the occurrence seems to be far underestimated, measured by a scarcity of scientific studies. In contrast, lameness and dirtiness are commonly used as welfare indicators in scientific studies. Further, these three indicators were also chosen for practical reasons as one observer scored lameness, dirtiness, and abomasal disorders.
  • We think it is appropriate to give the reader in the introduction the background knowledge of the different scoring systems instead of citing the literature in the material section. Therefore, we would like to keep it as it is.

In the methods part you are not exactly describing all the methods and scores you used. You are describing them in table 1, which is placed in the results part. But this table, as well as Figure 1 belong clearly to the methods part! In the last column of table 1 you show the distribution of the different scores in your observations and I guess you thought that those are results. This is true, but they were actually used to calculate the statistical models which are also described in the methods part and the greatest part of table 1 is clearly on methods. Therefore, I strongly recommend to move Table 1 and Figure 1 to the methods part.

  • Thank you for your comment. We have now mentioned Figure 1 in the material part. Table 1 was already part of the material part (please see line 117): Animal-related meta-data (independent variables) such as housing type, breed, sex and production system (see detail description in Table 1 and Figure 1) ….“

In the methods part you are always talking about the “cleanliness score”. But in the results part you are showing “contamination levels” (Figure 2). This should be consistent throughout the paper.

  • Thank you very much comment. We have now replaced „contamination levels with „cleanliness level“ (please see line 230).

Statistics: did you calculate any interactions? I wondered whether there could have been an interaction between breed and housing system.

  • We developed different types of models with different factor selections (as described in the material method) including intersection but the best accuracy of the model was without the correlated factors and without intersection caused by inhomogeneous distribution between both intersection factors such as for breed and housing systems.

The results part contains a lot of methods. Also lines 163 to 173 and lines 179 to186 should be moved to the methods part. On the other hand, there are aspects lacking in the results part, which appear only in the discussion part: lines 287 – 289 and lines 293 -295 describe interesting results (that transport- and waiting time is lower when farmers bring their animals to the slaughterhouse and that those parameters don’t have an impact on the observed welfare indicators). Those results have to be mentioned in the results part and not only in the discussion part.

  • Thank you for your comment. Lines 163-173 are in the material section: In both models the significance level was set to p < 0.05. The models were implemented using the packages 'ordinal', 'ggeffects', 'lme4', 'car', 'caret', 'lattice', 'ROCR', 'pROC' and 'LMERConvenienceFunctions' in the R (Version 4.0.5) statistical computing environment.

Lines  179-186 are in the result section and should be in our opinion part of the result section because in these lines we describe the prevalence for each factor such as:The majority of the analyzed slaughter cattle had no  lameness (99.27%; score 1 (n=409)). The frequency of the deep abomasal ulcers was low  (i.e. 7.78% (n=32; type 2) and 0.24% (n=1; type 3), respectively)) compared to cattle free of lesions with 47.81% (n=197; type 0)”

The conclusions part is too short. Please give your recommendations resulting from your study, here.

  • Thank you for your comment. Additionally, we have now included the following sentence in the conclusion section: In contrast to dirtiness and abomasal disorders prevalence, the determined lameness prevalence was very low. The husbandry of cattle was identified as a significant influencing factor for both the dirtiness and occurrence of abomasal disorders of slaughter cattle.

General remark on language: I found too many spelling and grammar mistakes (see below): those could have been avoided if a person with good English skills would have corrected the manuscript before submitting!

  • Many thanks for the numerous suggestions regarding language correction. We have now incorporated all of your suggestions (except for two; see below) throughout the paper and checked the paper carefully regarding the language again.

The two exceptions are:

  • Both „organic“ and „commercial“ are two different production systems in Austria (see Table 1) therefore we can not replace „commercial“ by „organic“.
  • NA means "not applicable" because it doesn't make sense to plot 412 farms IDs and ear tag numbers of all cattle in a histogram. We have now replaced NA in Tabel 1 with „Not applicable as a diagram“.
  • We have deleted the following sentence because we couldn't find a reference for this in the discussion section, as the reviewer's request: Another reason for observing the low lameness prevalence might be, that cattle might not exhibit lameness in stressful situations such as transport and in slaughter plants.